# Optimization of the Use of Hospital Beds as an Example of Improving the Functioning of Hospitals in Poland on the Basis of the Provincial Clinical Hospital No. 1 in Rzeszow

**DOI:** 10.3390/ijerph19095349

**Published:** 2022-04-28

**Authors:** Sławomir Porada, Katarzyna Sygit, Grażyna Hejda, Małgorzata Nagórska

**Affiliations:** 1Institute of Health Sciences, Medical College of Rzeszow University, 35-315 Rzeszow, Poland; ghej@wp.pl; 2Faculty of Health Science, Calisia University, 62-800 Kalisz, Poland; k.sygit@akademiakaliska.edu.pl; 3Institute of Medical Sciences, Medical College of Rzeszow University, 35-315 Rzeszow, Poland; mnagorska@ur.edu.pl

**Keywords:** health care, hospital bed occupancy, cost effectiveness, hospital restructuring, optimalization

## Abstract

An efficient health care system combines maximum accessibility with high-quality treatments, as well as cost optimization of individual health care facilities throughout the entire system. In hospitals, the critical element is the number of beds within individual wards, which generates costs and, at the same time, affects the capacity to serve patients. The aim of this article is to discuss the restructuring and optimization of hospital bed occupancy in a healthcare facility in the Podkarpackie voivodeship. The analysis covers the years 1999–2018. In the indicated period, the analyzed healthcare institution restructured the number of beds based on a forecast of the demand for services, which resulted in positive cost effects, without limiting patients’ access to diagnostic and therapeutic care. The analyzed facility took part in a common trend of optimizing cost-effectiveness and efficiency of hospital operations in Poland.

## 1. Introduction

Hospitals are a crucial element of the social infrastructure of any modern country [1,2]. Their efficient functioning requires both fixed and variable costs (in particular, investment costs [3]. In regard to fixed costs, it is important to constantly maintain a certain number of hospital beds in wards, as too many would generate excessive costs, and their effective occupancy would be too low. On the other hand, too few beds would lead to a significant risk of lack of access to diagnostics and treatment, which could reduce effectiveness of the health system [4,5]. Therefore, healthcare managers look for ways to optimize and restructure the occupancy of hospital beds [6,7].

Restructuring is a process commonly used in the management of organizations. It is a sequence of significant changes that are intended to lead to the improvement of the facility’s management efficiency [8]. The essence of the reorganization is a sudden change in assets, capital, or organizational structure that includes many different operations, as a result of which it is possible to maximize the positive effects of facility’s operations and minimize the negative effects of its weaknesses [9]. Hence, reorganization is aimed at optimizing the operations of an organization [10]. Process optimization, on the other hand, is aimed at the elimination of losses that lead to low operational or economic efficiency [11]. It is a method of implementing the optimal solution, using a specific set of qualitative criteria [12]. Optimizing operations of an entity is related to the concept of continuous improvement [13,14].

The idea of restructuring and optimization in the healthcare sector is consistent with the definitions presented here [15]. The qualitative difference in relation to changes of this type in business are different goals, escape from the narrow logic of profit, an attempt to balance the social aspect (the availability of health care) and economic variables, which relates primarily to the scale of costs [16,17,18,19]. Two distinctive perspectives should be noted: that of a health care manager, who manages the hospital, strives to rationalize costs and maximize revenues; and that of public facility, which implements health policy, looks for an increase in access to procedures and savings in terms of funding for health care. Reconciling these two perspectives while focusing on the priority—which is the patients’ wellbeing—is a systemic challenge for health care in Poland as well [17,18].

Optimization and restructuring are very important categories related to hospital operations in Poland and other developed countries, where new methods of using public funds are looked for, so that the health system can achieve its goals at the highest possible level, and at the same time will not generate excessive costs, which pose a serious problem due to budget restrictions [19,20]. Thus, activities focused on the number of hospital beds are an important aspect in optimizing the cost-effectiveness and efficiency of hospital operations in Poland [21,22].

The aim of this article is to discuss the restructuring and optimization of hospital bed occupancy in a selected healthcare facility in the Podkarpackie voivodeship. The analysis concerns a healthcare facility which has the status of a provincial hospital. It operates in the Podkarpackie voivodeship and is run by the regional self-government. The hospital serves patients from Rzeszow—the capital of the voivodeship—as well as residents of the entire Podkarpacie region. Its main statutory task is to provide in-patient and round-the-clock health services as part of its specialization, including hospital services. Part of its offer is diagnostic activity aimed at identifying the state of health and determining further treatment. It also runs its own research. In the Section 2, the authors focus on the material and research method, presenting the data in order to discuss the issue of restructuring and optimizing the occupancy of hospital beds. The Section 3 presents the results of the authors’ own research carried out using the previously indicated methodology in a selected hospital from the Podkarpackie voivodeship. The key element of the study is the discussion of the results presented in the Section 4 of the article.

## 2. Materials and Methods

The study is based on the analysis of documents. The analysis encompassed documentation obtained from the analyzed provincial hospital from the Podkarpackie voivodeship. Internal annual reports were used, which presented data such as number of beds, person-days that indicate the length of the patient’s stay in the hospital, contract size, and head count. Additionally, an important source of information was the remedial program, which was developed in 2017; it summarized optimization and restructuring activities undertaken since the late 1990s, as well as indicated additional future directions for the facility.

The analysis period was a time of intensive activities related to the optimization and restructuring of the number of beds in that institution. An important factor in the analysis is the fact that in 2017 the hospital was merged with the regional center for the treatment of lung diseases. Thus, the results for 2017–2018 relate to an institution that included new pulmonary clinics, which translated into an increase in the number of beds, as well as higher headcount and more ward equipment. The data are presented in charts; for some of them, indicators were calculated to show the change in the availability of beds. In addition to the information for that hospital, national statistics for Poland will be presented, which allows for setting the figures of the analyzed institution in the context of national hospital landscape.

## 3. Results

The key hospital resource that enables the provision of healthcare services is beds; their number in the analyzed period is presented in Figure 1.

The analyzed healthcare institution gradually limited the number of beds available to patients. The starting point for the analyzed period was 696 beds in 1999. The performed restructuring operations were based on the assumption that bed occupancy was below expectations, which led to staffing costs, as well as constant expenditure on the maintenance of equipment and the ward space. Hence, there was a visible decrease between 1999 and 2014, when the minimum level of 544 beds was reached. It should be added that there was no uniform downward trend, and the number of beds in the hospital fluctuated. There were cyclical declines (the largest was 7% between 1999 and 2000) and minimal increases (the highest was less than 2% in 2001 and 2002). In general, however, there was a downward trend in the number of beds in the hospital and between 1999 and 2014 it decreased by 152, or 21.83%.

In 2017–2018, a very significant increase in the number of beds in the ward may be observed. It increased by about 30% compared to 2016, but this was not the result of a significant expansion of its own wards, but of the above-mentioned merger of the hospital with a regional lung clinic. Table 1 shows the structure of the number of beds in the institution.

The main units in the examined institution were the following clinics: gynecology and obstetrics, tuberculosis and pulmonary diseases, as well as general and oncological surgery and radiotherapy. In all of these, the number of beds was a minimum of 50. This healthcare facility also had small, multi-bed anesthesia units (separate for the hospital and the pulmonary disease center, where the unit had the status of a ward). General data for Poland, shown in Figure 2, are also presented.

The nationwide trend was different than in the presented hospital from the Podkarpacie region. Between 2005 and 2018, there was a slight increase in the number of hospital beds. At the same time, if one compares the minimum and maximum levels for the analyzed period, there was a change from 175,000 in 2007 to 188,100 in 2014. The highest increases were observed between 2007 and 2008, and 2011 and 2012 (4.9% and 4.5%, respectively), while the decreases were smaller: a maximum of 1.9% between 2017 and 2018. In the latter case, it was the effect of introducing the so-called Hospital Network Act, which restructured the national hospital system. As a result of these changes, the parameter of the number of beds in general hospitals per 10,000 population in 2018 decreased from 48.2% (the record level was 49.0% in 2012) to 47.3%, which is an indicator close to the baseline level (47.0% in 2005; the minimum was 45.9% in 2007). It is also worth presenting the data by voivodeships, as seen in Table 2.

Changes in the analyzed facility did not correspond to the results for the entire Podkarpackie voivodeship. There was a significant increase in the number of hospital beds per 10,000 people. It spiked from 42.27 to 46.66, which was 4.39 beds per 10,000 population. This upward trend also applied to other regions of the country, as there was a decrease in the number of beds per 10,000 inhabitants in only 6 out of 17 voivodeships. In addition to the number of beds, attention should be paid to their occupancy within the institution, which allows for better identification of the actual need for them. The data are presented in Figure 3.

Statistics for the analyzed institution show that bed occupancy varied in the analyzed period from 69% in 2018 to 93% in 2000. After 2000, bed occupancy gradually decreased and reached the minimum in 2010 (72%). There was a very clear correlation between the number of beds and their occupancy (Pearson’s correlation coefficient at the level of 0.788 for the period 1999–2010). The level of both indicators gradually decreased, which indicates that the demand for healthcare services was falling. Thus, despite the decreasing number of beds, the occupancy rate was also characterized by a downward trend. After 2010, there was a gradual increase in the figures—to the level of 85% in 2016 (the last year before the merger with the lung disease clinic). In 2018, the lowest figure for the already enlarged facility was recorded in the entire analyzed period, which may also be related to the record number of beds for the period 1999–2018. Table 3 compares the national data and the data for the analyzed institution (the authors took 2015 as an example in the analyzed period of 1999–2018, because the most reliable and reliable data were obtained from this year).

The comparison of the efficiency of hospital bed occupancy in the analyzed institution showed that usually the occupancy rate in a regional hospital was higher than in the national average. This was especially visible in clinics such as: clinical oncology, otolaryngology, but also gynecology and obstetrics, where the indicators exceeded 90% and were significantly higher than the national indicators (this did not apply only to the hepatology clinic, where the data for Poland and the analyzed hospital were similar). The general psychiatric ward was a special case: the occupancy significantly exceeded the number of beds. Indicators below the national average existed in six wards, two of which were an anesthesiology and intensive care unit/clinic. Therefore, it is reasonable to say that the restructuring of the number of beds in the analyzed hospital contributed to the relatively high efficiency of their use, especially when the data for the analyzed institution and the national average are compared. Figure 4 shows person-days in the analyzed hospital.

The results show that the number of person-days in the analyzed facility decreased in 1999–2018. Between 1999 and 2016, there was a decrease from 218.4 to 170.0 (the minimum level was 145.1 person-days in 2010). By shortening the length of stay of patients, it was possible to maximize the use of hospital beds. Table 4 shows the average length of stay of patients by hospital wards.

The average length of patient stays in the hospital varied by clinic. Patients stayed the longest in the following clinics: general psychiatric, radiotherapy, hepatology, anesthesiology and intensive care, where the average length exceeded 10 days. For the lung disease clinic, this level was exceeded by the tuberculosis and lung disease clinic. The comparison with the regional and national data showed that the analyzed hospital was below the average person-days for, among others, neurology, clinical oncology, and nephrology, but when it came to psychiatry, tuberculosis and gynecology and obstetrics, it provided more person-days of care. This made it necessary to optimize the number of hospital beds not only according to the demand, but also to the specificity of providing medical procedures in a given facility. Figure 5 shows the contract sum and the amount per bed within the analyzed institution.

The contract amount for the hospital grew successively between 2002 and 2018. Attention should be paid to 2009, when the amount increased significantly and had to be adjusted in the following years. Disregarding that one-off excessive increase, the value of the contract increased from PLN 69.45 to PLN 158.68 million before the merger of the two entities, and to PLN 248.22 million in 2018. This also translated into a changed amount per hospital bed. In 2002, it was PLN 103,500 per bed, while in 2016 it was PLN 289,600. The following years saw a further increase to the level of PLN 345,700. Thus, there is a parallel optimization of the number of beds and an improvement in their financing at the systemic level. In Figure 6, the head count is analyzed in a similar manner.

In the analyzed period, an increase in employment of both nursing and medical positions was observed. Between 2002 and 2016, the number of nurses increased from 419 to 507; for doctors, there was an increase from 174 to 189. Further increases resulted from the merger with the lung disease clinic. Taking into account the fact that head count increased, and the number of beds was reduced, the ratio of nurses per bed had to increase. There was a change from 0.62 to 0.93 in 2016 and 0.88 in 2018 after the merger with the pulmonary clinics, while for doctors it was 0.26 in 2002 and 0.34 in both 2016 and 2018. The main findings of the research section are referred to in the discussion.

## 4. Discussion

Actions taken by the managers of the analyzed healthcare facility consisted of reducing the number of beds in the entire hospital. This meant simultaneously reducing their number in clinics where they were unnecessary, but also maintaining or increasing their number, if there was such a demand on the local medical market for a given procedure. Thus, it was possible to optimize the number of beds. That policy was consistent with findings by K. Wielicka, H. Dźwigoł and A. Męczyńska, who indicated that in the hospitals they’d studied, the number of beds was reduced by 8%, which translated into favorable cost effects [23]. H. Kromolowski, who examined a large provincial hospital in Łodź, suggested that reducing the number of beds by 10 to 20%—depending on the ward—would help achieve restructuring goals [23,24]. At the same time, it is not possible to provide quantitative values that constitute a model for the bed optimization, as this depends on the demand and the specificity of each facility.

The task in the course of optimization was to ensure that the rate of the hospital bed occupancy was appropriate. When it is too low, waste of resources occurs (the maintenance of beds requires funds). Naturally, in healthcare, it is necessary to adopt a long perspective, as it cannot be concluded that the bed occupancy is low in a period of several weeks or months, when the effect of a seasonal reduction in the number of medical services of a given type may occur. Therefore, it is important to determine, for example, the indicators of the share of the bed occupancy in the costs, or the relative increment of the occupancy rate. The psychiatric ward of the analyzed hospital needs to be mentioned: its number of beds was insufficient, which was visible by analyzing the bed occupancy rate—which exceeded 120%. This indicates that the number of beds in this clinic was insufficient, which was part of a systemic problem in the context of Polish child psychiatry [25], but which also challenged the treatment of adults. As Y. Pelto-Piri wrote on Swedish experiences, proper functioning of mental health centers requires, inter alia, proper equipment and the provision of competent staff focused on patients. Otherwise, the achieved results should be described as ‘unsatisfactory’. The availability of beds is a necessary condition for properly organized psychiatric treatment [26].

Returning to the issue of reducing the number of hospital beds, it is worth noting that it is a common practice in Poland to limit hospitals and hospital beds, as well as shifting some tasks from in-patient care to out-patient care, including home care [26,27]. It is worth recalling that the presented data for Poland showed that the number of beds increased in recent years [27]. This does not necessarily mean, however, that the number is sufficient to meet the needs of the society. Conversely, the removal of some of the beds in the analyzed hospital does not have to translate into limitation of its activities. On the contrary, it may be a mechanism leading to a better functioning of the hospital as a whole—as the costs incurred can be shifted to areas where the funds would be used more effectively. The ability of a facility to provide high-quality medical services and to respond to demand is of fundamental importance [28,29].

From the perspective of this analysis, but also based on a broad system analysis, it is necessary to improve the availability of health services, increase the quality of healthcare and, at the same time, optimize costs. Often in the literature one may find a justified view regarding the systemic error, in which the gradual reduction in the number of beds leads to the actual limitation of the availability of medical services in the area served by the hospital. This results in short-term positive effects in the profit and loss statement of the facility, but at the same time has negative consequences for the local community [30]. This is by no means an exclusively Polish problem. As noted by Goldwasser and his team, who analyzed the challenges faced by the Brazilian health service, reducing the number of beds is a common mechanism that limits the effectiveness of health care and the availability of procedures for patients [31]. Thus, mere limitation of the hospital beds does not have to be interpreted as problematic, provided that it is a mechanism for improving facility’s operations, seeking higher efficiency—and not just a simple cost-cutting practice.

Regarding the analyzed hospital in Rzeszow, it may be concluded that the aim of the restructuring program was unambiguous: to maintain the high availability of services, while reducing costs. The number of hospital beds was only one of its elements. The most important challenges included, for example, calculating the demand for health services and forecasting the demand. The existence of effective forecasting translated into proper planning of the hospital’s work and, as a result, into the level of funds that were transferred by taxpayers [32]. The number of hospital beds in a ward is a derivative of forecasting activities; it is a part of a wider set of data that together allows the management of a healthcare facility to make decisions.

It should be noted that after the introduction of the so-called Hospital Network Act (or rather an Act amending the Act on Health Care Benefits Financed from Public Funds) [33], there was a global change in the number of beds in Polish hospitals [34]. The act itself was intended to increase the systemic effectiveness of healthcare institutions. Layers of safety mechanisms were created, and financing mechanisms were remodeled, which required structural changes in many institutions [35,36]. In the analyzed hospital, the act did not contribute to deep internal transformations in terms of the number of beds, mostly because it had been modified earlier. Nevertheless, the general data for Poland shows a slight decrease in the number of hospital beds, despite the fact that earlier years led to their increase [37,38].

Regarding bed optimization, one may also focus on the relationship between them and the level of employment. As our own study showed, there was an increase in the number of nursing and medical personnel per bed. It seems that combining such data is very important, as the number of beds is not the key factor for patient support. What is more important is the specialist staff that carries out diagnostic and treatment activities for patients occupying those beds. Labor costs in health care are of key importance and, as shown in the literature, they often amount to over 50% of the facility’s budget [39]. Therefore, the increased headcount in the hospital in Rzeszow was positive, but it should be noted that the number of doctors and nurses employed as contract staff increased significantly. Flexible forms of employment are eagerly chosen in Polish hospital care, as pointed out by, for example, M. Olkiewicz, because they provide cost benefits [40]. Nevertheless, they are also related to some problems, such as overworked staff due to lack of legal working time limits for contractual staff. Thus, improving the staffing rate per bed is associated with an increase in contract workforce, which may be a problem for the hospital in the long run.

The relationship between hospital beds and a contract for the hospital in Rzeszow was also investigated. It was found that significantly more funds were allocated per bed, which is a positive change resulting from the general increase in health care financing in Poland. The already-mentioned Porębski indicated that the size of the contract depended on the stability of the hospital, and the extent to which the resources, including beds, could be used effectively by the facility [34]. Many hospitals are struggling with contract amounts that are too low, which reduces budgetary stability. Sometimes this is the fault of unfavorable structures of procedure financing by the National Health Fund, but it may also be due to the fact that the institutions’ processes are not appropriate, and some restructuring is necessary [23,41]. Regardless of these considerations, higher funding per bed translates into better operational possibilities, although it is necessary to recall the importance of the inflation factor and the general increase in the prices of medical procedures, which results from, e.g., progress in the field of pharmacology and medical technology.

Fundamentally important for optimization of hospitals is the proper balance between the operational capabilities of the facility and costs (especially fixed ones). As Y. Gu and Q. Luo wrote, the key is to parameterize such factors as: the economic cost of a bed, the number of hospital employees, as well as the number of patients per bed [42]. Calculations should be used as a basis for any improvement activities, and changes in the number of beds must be based on facts. Meanwhile, J. Nguyen et al. wrote about algorithms that help to balance costs and maintain access to medical procedures, which affects the broadly understood effectiveness of a facility (and not only its cost-effectiveness) [43]. It seems that the managers of the analyzed hospital understood optimization not only as elimination of ‘deficit’ beds, but as a chance to improve the efficiency of the facility by matching the number of beds to the actual demand in the region.

Restructuring activities in the hospital were based on the data from IT systems. Today it is a standard method of improving the functioning of both individual health care facilities and the entire health care systems [43]. Modern digital solutions, such as artificial intelligence and data mining processes, along with data-driven mathematical modeling, may determine to what extent the current number of beds is optimal from the perspective of the healthcare facility. Therefore, the future will enable much more effective optimization of all hospital processes, as the amount of data and the processing mechanisms will help to create simulations used in decision-making. Their flexibility will make it easier for hospitals to adapt to the changing demand for individual procedures [44].

The presented analysis of hospital bed stock in key western European countries in terms of health care development, i.e., France, Germany, Italy, and Spain, before the COVID-19 pandemic shows that all countries saw a reduction in hospital beds from 2010 to 2017. This tendency resulted from organizational and economic factors. The most significant rationale was the reduction of hospital benefits and their replacement with day-care treatments and the decrease in the number of hospital beds resulting from the transition to home and territorial care. It ought to be noted that the achieved economic, as well as qualitative and organizational, effectiveness in the form of the reduction of hospital beds was faced with the COVID-19 pandemic situation and the enormous demand for hospital services. The shortage of hospital beds in the health systems of these countries, in particular in the regions most affected by COVID-19, put enormous pressure on the hospitals operating in these countries [45,46,47].

Finally, it should be mentioned that the issue of restructuring hospitals and hospital beds is not a typically Polish phenomenon; neither is it related to post-transformation economies. Rather, it is an element typical of any health systems that experience pressure to reduce costs [43].

The facility in Rzeszow also struggled with the problem described by Scandinavian authors, but it was possible to optimize both the number of beds in the delivery ward and to increase the importance of the Gynecology and Obstetrics clinic on the local healthcare market. Thus, the challenges faced by Polish healthcare institutions are part of a wider problem faced by hospitals around the world [47,48,49,50,51]. Local financial and legal conditions, as well as the existing demand for health services, are the framework that sets the specific background for the restructuring and optimization of medical facilities, including hospital beds.

The perspective of the health system is an important context, as legal and institutional framework is created at the national level; financing of medical services is budgeted at the national level as well. Restructuring and optimization must take into account local and regional specificities (if they concern large provincial hospitals). Therefore, the analyzed facility conducted multifaceted continuous improvement aimed at process and cost efficiency. The most important aspect was the number of hospital beds and their occupancy rate. The main goal was to plan for such resource level that would allow to maximize the availability and quality of treatment for the patient, while obtaining the correct level of fixed costs related to access to hospital beds.

## 5. Conclusions

As a result of the analyses, it was found that the number of beds in the analyzed period decreased, which was a symptom of restructuring activities aimed at improving the efficiency of using each place. It is important that the demand for health services in the surveyed hospital decreased in the analyzed period, and a very significant increase in the number of medical procedures is a derivative of a change in the method of their calculation. It was also found that the occupancy rate of the beds was relatively constant, which indicates that measures to reduce the number of beds were necessary as otherwise the occupancy rate would drop dramatically. Another important conclusion is the indicated increase in the financing of medical services in terms of the entire hospital’s finances, which has led to a significant increase in funds per one hospital bed, which is a positive change in the global context. It should also be pointed out the increase in medical personnel, in particular doctors and nurses per bed, which is a positive aspect as the key importance for patient support is not only the number of beds but, above all, the staffing of medical personnel.

The presented conclusions are intended to emphasize the important role of optimizing the number of hospital beds in the perspective of the functioning of the hospital system. The recommendations from the study are addressed to healthcare entities restructuring the number of beds, with particular emphasis on the issue of improving the quality and efficiency of their use.

In the future, it would be worth expanding research in other units with regard to the optimization and effectiveness of hospitals in terms of the use of: medical personnel, hospital infrastructure, public funds, changing demand for medical services.

## 6. Limitations

The present study on bed optimization in hospitals is a preliminary analysis of optimization and restructuring activities in Polish hospitals over the past twenty years. Further research on the issues of hospital reform will allow for a broader analysis and drawing constructive conclusions as to the optimization and restructuring measures in terms of the functioning of modern hospital units. Drawing the conclusions will lead to working out a model scenario of inpatient hospital care functioning while accounting for quality, financial, and health factors.

## Figures and Tables

**Figure 1 ijerph-19-05349-f001:**
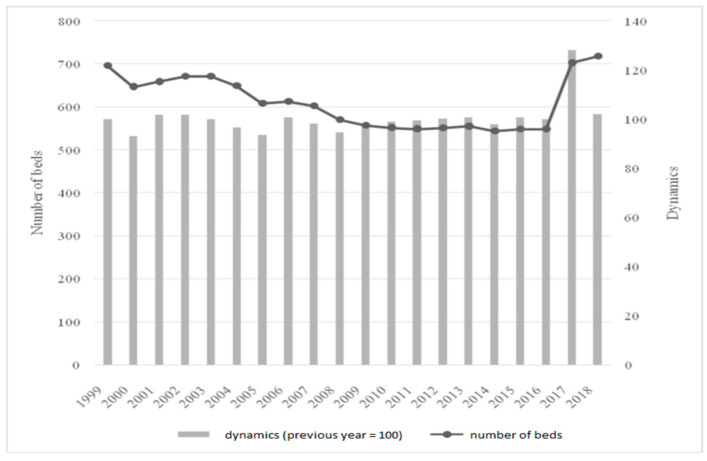
Number of beds available in the analyzed hospital in 1999–2018.

**Figure 2 ijerph-19-05349-f002:**
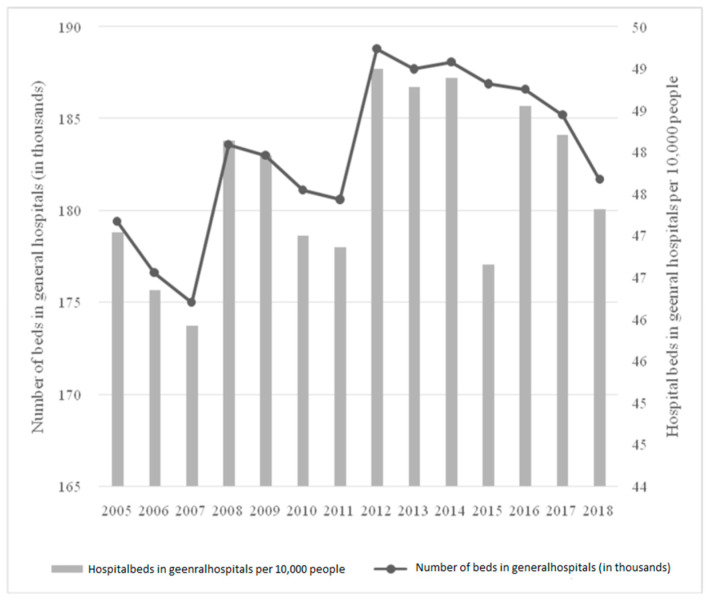
Hospital beds in Poland in 2005–2018 per 10,000 people and number of beds in general hospitals.

**Figure 3 ijerph-19-05349-f003:**
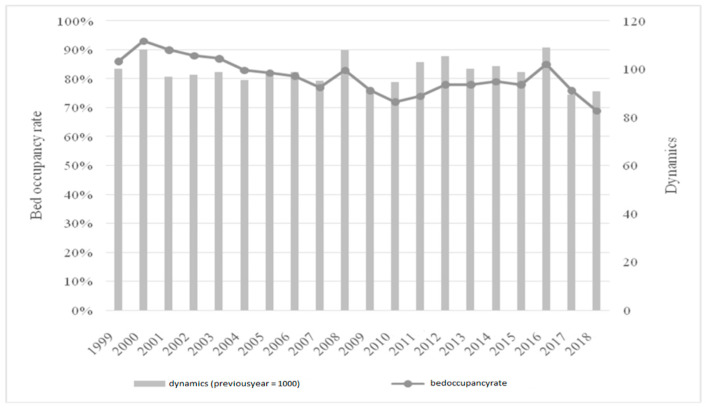
Bed occupancy in the analyzed hospital in 1999–2018.

**Figure 4 ijerph-19-05349-f004:**
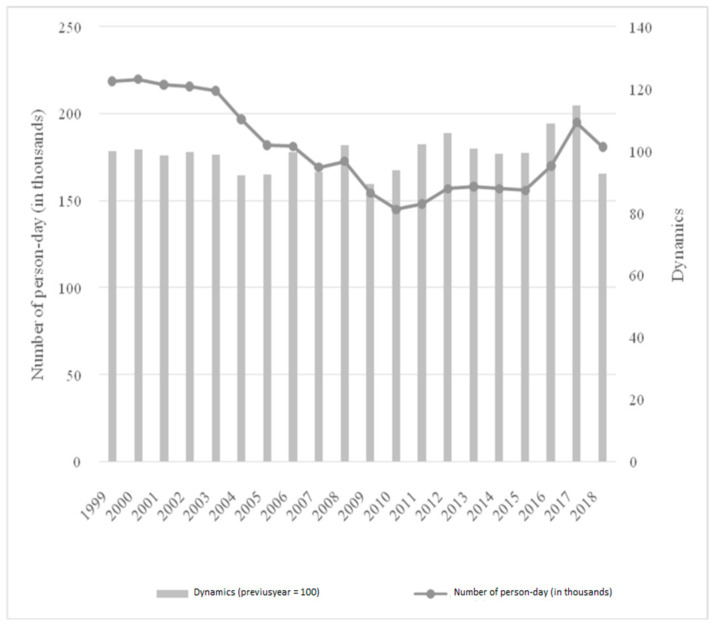
Person-days of patients in the analyzed hospital in 1999–2018.

**Figure 5 ijerph-19-05349-f005:**
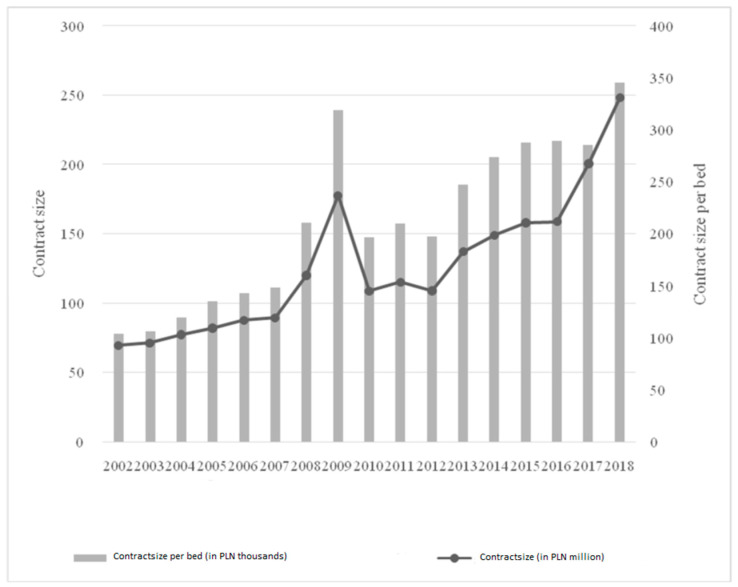
Contract of the analyzed hospital and the amount per hospital bed in the analyzed health care institution.

**Figure 6 ijerph-19-05349-f006:**
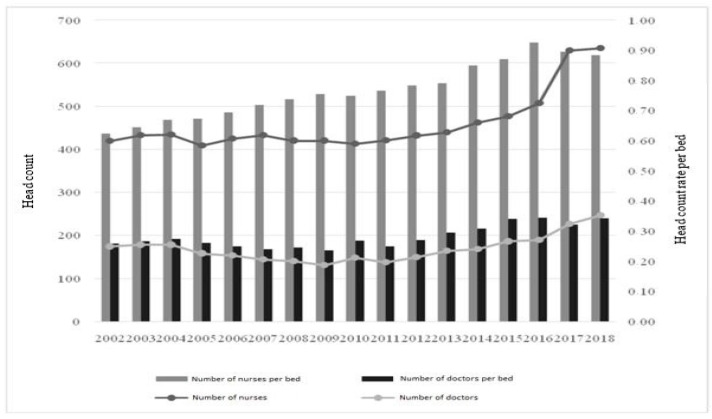
Medical and nursing staff of the analyzed hospital per hospital bed in the analyzed institution.

**Table 1 ijerph-19-05349-t001:** Number of beds in the clinics of the analyzed hospital in 2017 (excluding day beds).

Clinic	Number of Beds	% of Beds
Anesthesiology and Intensive Care	8	1.08%
Clinical Oncology	25	3.36%
General and Oncological Surgery	50	6.73%
Maxillofacial Surgery	17	2.29%
Gastroenterology and Hepatology	43	5.79%
Dermatology	24	3.23%
Gynecology and Obstetrics	86	11.57%
Oncological Gynecology	35	4.71%
Hepatology	24	3.23%
Nephrology	18	2.42%
Ophthalmology	35	4.71%
Otolaryngology	38	5.11%
General Psychiatry	30	4.04%
Radiotherapy	50	6.73%
Urology	42	5.65%
Neurology	23	3.10%
Newborns with Intensive Care Unit	40	5.38%
Pulmonology with Allergology *	30	4.04%
Pulmonology and Chemotherapy *	26	3.50%
Thoracic Surgery *	27	3.63%
Tuberculosis and Lung Disease *	68	9.15%
Anesthesiology and Intensive Care Unit *	4	0.54%

* Lung disease clinics.

**Table 2 ijerph-19-05349-t002:** Beds in general hospitals per 10,000 people in Poland, broken down by voivodeships, in 2005–2018.

Voivodeship	Years				
	2005	2010	2015	2018	2005/2018
Lower Silesia	48.98	48.42	51.10	50.53	+1.55
Kuyavian-Pomeranian	43.59	42.97	47.20	46.16	+2.57
Lubelskie	50.80	51.82	52.84	51.89	+1.09
Lubuskie	42.90	40.96	43.24	42.18	−0.72
Lodzkie	53.20	53.23	52.07	50.79	−2.41
Lesser Poland	42.56	42.78	44.06	43.68	+1.12
Masovian	45.59	45.92	48.47	47.69	+2.10
Opolskie	39.76	43.13	46.22	44.78	+5.02
Podkarpackie	42.27	44.91	48.17	46.66	+4.39
Podlaskie	50.92	49.61	49.91	49.83	−1.09
Pomeranian	38.89	38.27	41.19	39.00	+0.11
Silesian	57.12	56.07	55.85	54.46	−2.66
Świętokrzyskie	44.96	50.25	50.22	46.08	+1.12
Warmian-Masurian	42.24	41.17	46.32	44.86	+2.62
Greater Poland	46.55	45.36	45.34	43.11	−3.44
West Pomeranian	46.23	45.17	48.34	46.10	−0.13

**Table 3 ijerph-19-05349-t003:** Bed occupancy rate in the analyzed institution and in Poland in 2015.

Clinic	Analyzed Hospital	Poland
Anesthesiology and Intensive Care	60.8%	68.9%
Clinical Oncology	96.9%	71.4%
General and Oncological Surgery	71.7%	no data
Maxillofacial Surgery	83.9%	46.9%
Gastroenterology and Hepatology	78.2%	no data
Dermatology	80.1%	63.7%
Gynecology and Obstetrics	90.6%	58.0%
Oncological Gynecology	53.8%	no data
Hepatology	90.7%	89.2%
Nephrology	70.8%	75.5%
Ophthalmology	65.8%	43.0%
Otolaryngology	96.0%	52.3%
General Psychiatry	122.8%	92.7%
Radiotherapy	68.7%	no data
Urology	58.3%	63.1%
Neurology	49.3%	76.0%
Newborns with Intensive Care Unit	68.0%	50.2%
Pulmonology with Allergology *	67.3%	no data
Pulmonology and Chemotherapy *	73.1%	no data
Thoracic surgery *	60.9%	no data
Tuberculosis and Lung Disease *	65.6%	70.9%
Anesthesiology and Intensive Care Unit *	49.3%	68.9%

* Lung disease clinics.

**Table 4 ijerph-19-05349-t004:** The average length of stay of a patient in the clinics of the analyzed hospital, compared to the data for the Podkarpackie voivodeship and Poland in general (comparison for 2015).

Clinic	Hospital	Region	Poland
Anesthesiology and Intensive Care	11.1	11.7	8.5
Clinical Oncology	3.6	4.0	4.3
General and Oncological Surgery	6.0	no data	no data
Maxillofacial Surgery	3.4	3.4	3.6
Gastroenterology and Hepatology	3.7	no data	no data
Dermatology	8.3	8.0	6.1
Gynecology and Obstetrics	4.6	3.3	3.5
Oncological Gynecology	3.9	no data	no data
Hepatology	12.1	6.8	6.2
Nephrology	5.2	9.0	6.0
Ophthalmology	1.6	1.5	1.9
Otolaryngology	3.1	3.0	3.0
General Psychiatry	36.6	28.2	28.0
Radiotherapy	14.9	no data	no data
Urology	5.3	3.6	3.3
Neurology	4.1	6.0	6.8
Newborns	5.9	4.0	4.5
Pulmonology with Allergology *	6.6	no data	no data
Pulmonology and Chemotherapy *	5.6	no data	no data
Thoracic Surgery *	7.1	no data	no data
Tuberculosis and Lung Disease *	10.1	9.7	8.3
Anesthesiology and Intensive Care *	11.0	11.7	8.5

* Lung disease clinics.

## Data Availability

Not applicable.

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
