# Peer review of "Optimization of the Use of Hospital Beds as an Example of Improving the Functioning of Hospitals in Poland on the Basis of the Provincial Clinical Hospital No. 1 in Rzeszow"

_ijerph, 2022, doi:10.3390/ijerph19095349_

Round 1

Reviewer 1 Report

General comments:

The study is too limited as only one institution. Therefore, the results do not make a relevant contribution to the knowledge base in the field.

Detailed comments:

Title:

  • The title of the article ‘Optimizing the use of hospital beds as an example of improving hospital performance in Poland’ does not indicate that the analysis covers only one hospital in Podkarpackie voivodship.

Keywords:

  • Keywords should be selected according to Index Medicus Medical Subject Headings – MeSH, for example: ‘Hospital Restructuring’

Introduction:

  • Repetitions: for example: The analysis covers the years 1999-2018 appears twice in this section.
  • A redundant section in Introduction: ‘The article closes with conclusions that are to emphasize the role of restructuring and optimization of hospital bed occupancy as a sign of a common trend in the search for cost-effectiveness and efficiency of hospital operations’.

Material and Methods

  • Repetitions: for example: ‘Moreover, the data from the Polish Central Statistical Office (GUS) was used, 87 which helped present the context of the entire country’; ‘In addition to the information for that hospital, national statistics 96 for Poland will be presented, which allows for setting the figures of the analyzed institu-97 tion in the context of national hospital landscape’.

Results:

  • Figure 2. The title of the figure 2 indicates only the number of hospital beds and not the ratio of beds per 10,000 people which is also presented.
  • The choice of years for the data presented in the table 2 is incomprehensible - the analyzed material covers the period 1999-2018, while the table presents the values for the years: 2005, 2010, 2015, 2018.
  • Beds in general hospitals per 10,000 people in Poland (table 2) as an indicator is difficult to relate to the analyzed data from one selected hospital in Podkarackie voivodeship.
  • It is not understood why 2015 was selected for the presented in table 3 bed occupancy rate.
  • Incorrect table number indicated in the text: ‘Table 2 compares the national data 164 and the data for the analyzed institution’ (should be table 3 instead of table 2).

Discussion:

  • many important issues have been raised in the discussion section which do not always relate to the objectives of the analysis presented in this paper.
  • Little attention is paid to explaining what the results mean.

Conclusions:

  • The conclusions cannot be justified on the basis of the rest of the paper.

References:

  • The format of the references is not consistent with the recommended style
  • 25% of cited publications are 10 years old or more.
  • Wrong name of the institution: should be ‘Statistics Poland’ not ‘Polish Central Statistical Office’.
  • Typo errors: for example ‘Acton Health Care Benefits Financed from Public Funds’, ‘authorsfocus’.

Reviewer 2 Report

Thank you for giving me the opportunity to review this manuscript. There are few concerns which I would like to highlight before the paper could be considered for publication:

  1. My one major concern is that the analysis was purely descriptive without inferences being made. This made difficult to understand consistency and temporality of the patterns to be observed.
  2. Authors should at least explore few indicators that could have influence the trends statistically to be reported for more useful interpretations.
  3. The methodology does not suffice to be contextualized as a research a paper. Authors may need to elaborate further on the design, how document review was conducted, inclusion or exclusion criteria, any missing imputation strategies, analytical approach used, etc.
  4. If this is not possible, then authors should consider to change the type of their article such as a perspective article.
  5. In short, I was not able to capture what restructuring program the authors referred to? What was it done? How was it done? These could be elaborated further.
  6. Authors could come out with a theoretical concept framework on the scenario being studied. 
  7. Where is your ethics statement? It is not sufficient to just mention conformity with the guidelines of Helsinki Declaration.

Reviewer 3 Report

The article "Optimizing the use of hospital beds as an example of improving hospital performance in Poland" is an interesting and extensive study. It contains numerous tables and figures, thanks to which the presentation of results is clearer. However, it is worthwhile for authors to introduce a few additional aspects to the content of their work. First of all, it is worth considering the usefulness and relevance of the results of the presented research in an international context. In their present shape, they may be of primary interest to Polish readers. Moreover, it is necessary to add information about the limitations of the presented study.

Round 2

Reviewer 1 Report

Typo errors:

Line 22: opimalization

Line 211: hedcount

Repetitions:

Line 63-63: The analysis covers the years 1999-2018.

Line 86: The analysis covered the period 1999-2018,

Line 126: Figure 2. The title of the figure 2 indicates only the ratio of beds per 10,000 people and not the number of hospital beds which is also presented.

Line 163: It is not understood why 2015 was selected for the presented in table 3 bed occupancy rate.

Discussion section:

many important issues have been raised in the discussion section which do not always relate to the objectives of the analysis presented in this paper.

Conclusions:

The conclusions cannot be justified on the basis of the rest of the paper (the conclusions formulated are very general). The findings from the Results section were repeated in the Conclusions section, lack of recommendations resulting from the conducted analyses.

Reviewer 2 Report

Thank you for your responses and justifications. I accept the responses of the authors for this paper in its current form as a preliminary perspective. However, I would like to stress two important points that the paper needs to see if accepted for publication:

  1. The paper needs to be changed to a "Perspective" type article.
  2. Please mention in the ethics part a statement as justified in your responses: 

    Institutional approval was applied to the Bioethics Commitee of the University of Rzeszów and the management of the analyzed Hospital for the implementation of the study. The Bioethics Committee decided to waive ethical approval as this type of study did not require their consent.

    Thank you.
